# AN EMPIRICAL STUDY ON POST-PROCESSING METHODS FOR WORD EMBEDDINGS

## ABSTRACT

Word embeddings learnt from large corpora have been adopted in various applications in natural language processing and served as the general input representations to learning systems. Recently, a series of post-processing methods have been proposed to boost the performance of word embeddings on similarity comparison and analogy retrieval tasks, and some have been adapted to compose sentence representations. The general hypothesis behind these methods is that by enforcing the embedding space to be more isotropic, the similarity between words can be better expressed. We view these methods as an approach to shrink the covariance/gram matrix, which is estimated by learning word vectors, towards a scaled identity matrix. By optimising an objective in the semi-Riemannian manifold with Centralised Kernel Alignment (CKA), we are able to search for the optimal shrinkage parameter, and provide a post-processing method to smooth the spectrum of learnt word vectors which yields improved performance on downstream tasks.

## 1 INTRODUCTION

Distributed representations of words have been widely dominating the Natural Language Processing research area and other related research areas where discrete tokens in text are part of the learning systems. Fast learning algorithms are being proposed, criticised, and improved. Despite the fact that there exist various learning algorithms (Mikolov et al., 2013a; Pennington et al., 2014; Bojanowski et al., 2017) for producing high-quality distributed representations of words, the main objective is roughly the same, which is drawn from the Distributional Hypothesis (Harris, 1954). The algorithms assume that there is a smooth transition of meaning at the word-level, thus they learn to assign higher similarity for adjacent word vectors than those that are not adjacent.

The overwhelming success of distributed word vectors leads to subsequent questions and analyses on the information encoded in the learnt space. As the learning algorithms directly and only utilise the co-occurrence provided by large corpora, it is easy to hypothesise that the learnt vectors are correlated with the frequency of each word, which may not be relevant to the meaning of a word (Turney & Pantel, 2010) and might hurt the expressiveness of the learnt vectors. One can theorise the frequency-related components in the learnt vector space, and remove them (Arora et al., 2017; Mu et al., 2018; Ethayarajh, 2018). These post-processing methods, whilst very effective and appealing, are derived from heavy assumptions on the representation geometry and the similarity measure.

In this work we re-examine the problem of post-processing word vectors as a shrinkage estimation of the true/underlying oracle gram matrix of words, which is a rank-deficient matrix due to the existence of synonyms and antonyms. Constrained from the semi-Riemannian manifold (Benn & Tucker, 1987; Abraham et al., 1983) where positive semi-definite matrices, including gram matrices, exist, and Centralised Kernel Alignment (Cortes et al., 2012), we are able to define an objective to search for the optimal shrinkage parameter, also called mixing parameter, that is used to mix the estimated gram matrix and a predefined target matrix with the maximum similarity with the oracle matrix on the semi-Riemannian manifold. Our contribution can be considered as follows:

**1.** We define the post-processing on word vectors as a shrinkage estimation, in which the estimated gram matrix is calculated from the pretrained word embeddings, and the target matrix is a scaled identity matrix for smoothing the spectrum of the estimated one. The goal is to find the optimal mixing parameters to combine the estimated gram matrix with the target in the semi-Riemannian manifold that maximises the similarity between the combined matrix and the oracle one.

**2.** We choose to work with the CKA, as it is invariant to isotropic scaling and also rotation, which is a desired property as the angular distance (cosine similarity) between two word vectors is commonly used in downstream tasks for evaluating the quality of learnt vectors. Instead of directly deriving formulae from the CKA (Eq. 1), we start with the logarithm of CKA (Eq. 2), and then find an informative lower bound (Eq. 5). The remaining analysis is done on the lower bound.

**3.** A useful objective is derived to search for the optimal mixing parameter, and the performance on the evaluation tasks shows the effectiveness of our proposed method. Compared with two other methods, our approach shows that the shrinkage estimation on a sub-Riemannian manifold works better than that on an Euclidean manifold for word embedding matrices, and it provides an alternative way to smooth the spectrum by adjusting the power of each linearly independent component instead of removing top ones.

## 2 RELATED WORK

Several post-processing methods have been proposed for readjusting the learnt word vectors (Mu et al., 2018; Arora et al., 2017; Ethayarajh, 2018) to make the vector cluster more isotropic w.r.t. the origin such that cosine similarity or dot-product can better express the relationship between word pairs. A common practice is to remove top principal components of the learnt word embedding matrix as it has been shown that those directions are highly correlated with the occurrence. Despite the effectiveness and efficiency of this family of methods, the number of directions to remove and also the reason to remove are rather derived from empirical observations, and they don't provide a principled way to estimate these two factors. In our case, we derive a useful objective that one could take to search for the optimal mixing parameter on their own pretrained word vectors.

As described above, we view the post-processing method as a shrinkage estimation on the semi-Riemannian manifold. Recent advances in machine learning research have developed various ways to learn word vectors in hyperbolic space (Nickel & Kiela, 2017; Dhingra et al., 2018; Gulcehre et al., 2019). Although the Riemannian manifold is a specific type of hyperbolic geometry, our data samples on the manifold are gram matrices of words instead of word vectors themselves in previous work, , which makes our work different from previous ones. The word vectors themselves are still considered in Euclidean space for the simple computation of similarity between pairs, and the intensity of the shrinkage estimation is optimised to adjust the word vectors.

Our work is also related to previously proposed shrinkage estimation of covariance matrices (Ledoit & Wolf, 2004; Schäfer & Strimmer, 2005). These methods are practically useful when the number of variables to be estimated is much more than that of the samples available. However, the mixing operation between the estimated covariance matrix and the target matrix is a weighted sum, and $l_2$ distance is used to measure the difference between the true/underlying covariance matrix and the mixed covariance matrix. As known, covariance matrices are positive semi-definite, thus they lie on a semi-Riemannian manifold with a manifold-observing distance measure. A reasonable choice is to conduct the shrinkage estimation on the semi-Riemannian manifold, and CKA is applied to measure the distance as rotations and istropic scaling should be ignored with measuring two sets of word vectors. The following sections will discuss our approach in details.

## 3 SHRINKAGE OF GRAM MATRIX

Learning word vectors can be viewed as estimating the oracle gram matrix where the relationship of words is expressed, and the post-processing method we propose here aims to best recover the oracle gram matrix. Suppose there exists an ideal gram matrix $\boldsymbol{K} \in \mathbb{R}^{n \times n}$ where each entry represents the similarity of a word pair defined by a kernel function $k(w_i, w_j)$. Given the assumption of a gram matrix and the existence of polysemies, the oracle gram matrix $\boldsymbol{K}$ is positive semi-definite and its rank is between $0$ and $n$ and denoted as $k$.

A set of pretrained word vectors is provided by a previously proposed algorithm, and for simplicity, an embedding matrix $\boldsymbol{E} \in \mathbb{R}^{n \times d}$ is constructed in which each row is a word vector $\boldsymbol{v}_i$. The estimated gram matrix $\boldsymbol{K}' = \boldsymbol{E}\boldsymbol{E}^\top \in \mathbb{R}^{n \times n}$ has rank $d$ and $d \leq k$. It means that overall, the previously proposed algorithms give us a low-rank approximation of the oracle gram matrix $\boldsymbol{K}$.

The goal of shrinkage is to, after the estimated gram matrix $K'$ is available, find an optimal post-processing method to maximise the similarity between the oracle gram matrix $K$ and the estimated one $K'$ given simple assumptions about the ideal one without directly finding it. Therefore, a proper similarity measure that inherits and respects our assumptions is crucial.

## 3.1 CENTRALISED KERNEL ALIGNMENT (CKA)

The similarity measure between two sets of word vectors should be invariant to any *rotations* and also to *isotropic scaling*. A reasonable choice is the Centralised Kernel Alignment (CKA) (Cortes et al., 2012) defined below:

$$\rho(K_c, K_c') = \frac{\langle K_c, K_c' \rangle_F}{||K_c||_F ||K_c'||_F} \tag{1}$$

where $K_c = \left[ I - \frac{1}{n} \mathbf{1}\mathbf{1}^\top \right] K \left[ I - \frac{1}{n} \mathbf{1}\mathbf{1}^\top \right]$, and $\langle K_c, K_c' \rangle_F = \mathrm{Tr}(K_c K_c')$. For simplicity of the derivations below, we assume that the ideal gram matrix $K$ is centralised, and the estimated one $K'$ can be centralised easily by removing the mean of word vectors. In the following section, $K$ and $K'$ are used to denote centralised gram matrices.

As shown in Eq. 1, it is clear that $\rho(K, K')$ is invariant to any rotations and isotropic scaling, and doesn't suffer from the issues of $K$ and $K'$ being low-rank. The centralised kernel alignment has been recommended recently as a proper measure for comparing the similarity of features learnt by neural networks (Kornblith et al., 2019). As learning word vectors can be thought of as a one-layer neural network with linear activation function, $\rho(K, K')$ is a reasonable similarity measure between the ideal gram matrix and our shrunken one provided by learning algorithms. Given the Cauchy-Schwartz inequality and also the non-negativity of the Frobenius norm, $\rho(K, K') \in [0, 1]$.

Since both $K$ and $K'$ are positive semi-definite, their eigenvalues are denoted as $\{\lambda_{\sigma_1}, \lambda_{\sigma_2}, ..., \lambda_{\sigma_k}\}$ and $\{\lambda_{\nu_1}, \lambda_{\nu_2}, ..., \lambda_{\nu_d}\}$ respectively, and the eigenvalues of $KK'$ are denoted as $\{\lambda_1, \lambda_2, ..., \lambda_d\}$ as the rank is determined by the matrix with lower rank.[1] The log-transformation is conducted to simplify the derivation.

$$
\begin{aligned}
\log \rho(K, K') &= \log \mathrm{Tr}(KK') - \tfrac{1}{2} \log \mathrm{Tr}(KK^\top) - \tfrac{1}{2} \log \mathrm{Tr}(K'K'^\top) \\
&\geq \tfrac{1}{d} \log \left( \det(K) \det(K') \right) - \tfrac{1}{2} \log \mathrm{Tr}(KK^\top) - \tfrac{1}{2} \log \mathrm{Tr}(K'K'^\top) \\
&= \tfrac{1}{d} \sum_{i=1}^k \log \lambda_{\sigma_i} + \tfrac{1}{d} \sum_{i=1}^d \log \lambda_{\nu_i} - \tfrac{1}{2} \log \sum_{i=1}^k \lambda_{\sigma_i}^2 - \tfrac{1}{2} \log \sum_{i=1}^d \lambda_{\nu_i}^2
\end{aligned} \tag{2}
$$

where the lower bound is given by the AM$-$GM inequality, and the equality holds when all eigenvalues of $KK'$ are the same $\lambda_1 = \lambda_2 = ... = \lambda_d$.

## 3.2 SHRINKAGE OF GRAM MATRIX ON SEMI-RIEMANNIAN MANIFOLD

As the goal is to find a post-processing method that maximises the similarity $\rho(K, K')$, a widely adopted approach is to shrink the estimated gram matrix $K'$ towards the target matrix $T$ with a predefined structure. The target matrix $T$ is usually positive semi-definite and has same or higher rank than the estimated gram matrix. In our case, we assume $T$ is full rank as it simplifies equations.

Previous methods rely on a linear combination of $K'$ and $T$ in Euclidean space (Ledoit & Wolf, 2004; Schäfer & Strimmer, 2005). However, as gram matrices are positive semi-definite, they naturally lie on the semi-Riemannian manifold. Therefore, a suitable option for shrinkage is to move the estimated gram matrix $K'$ towards the target matrix $T$ on a semi-Riemannian manifold (Brenier, 1987), and the resulting matrix $Y$ is given by

$$Y = T^{1/2}(T^{-1/2}K'T^{-1/2})^\beta T^{1/2} \tag{3}$$

where $\beta \in [0, 1]$ is the mixing parameter that indicates the strength of the shrinkage, and $T^{1/2}$ is the square root of the matrix $T$. The objective is to find the optimal $\beta$ that maximises $\log \rho(K, K')$.

$$\beta^\star = \arg\max_\beta \log \rho(K, Y) = \arg\max_\beta \log \rho \left( K, T^{1/2}(T^{-1/2}K'T^{-1/2})^\beta T^{1/2} \right) \tag{4}$$

The objective defined in Eq. 4 is hard to work with, instead, we maximise the lower bound in Eq. 2 to find optimal mixing parameter $\beta^\star$. By plugging $Y$ into Eq. 2 and denoting $\lambda_{y_i}$ as the $i$-th eigenvalue of $Y$, we have:

$$\log \rho(K, Y) \geq \tfrac{1}{d} \sum_{i=1}^k \log \lambda_{\sigma_i} + \tfrac{1}{d} \sum_{i=1}^d \log \lambda_{y_i} - \tfrac{1}{2} \log \sum_{i=1}^k \lambda_{\sigma_i}^2 - \tfrac{1}{2} \log \sum_{i=1}^d \lambda_{y_i}^2 \tag{5}$$

---

[1] $\lambda_{\sigma_k} = \lambda_{\nu_d} = 0$ due to the centralisation.

### 3.3 Scaled Identity Matrix as Target for Shrinkage

Recent progress in analysing pretrained word vectors (Mu et al., 2018; Arora et al., 2017) recommended to make them isotropic by removing top principal components as they highly correlate with the frequency information of words, and it results in a more compressed eigenspectrum. In Euclidean space-based shrinkage methods (Anderson, 2004; Schäfer & Strimmer, 2005), the spectrum is also smoothed by mixing the eigenvalues with ones to balance the overestimated large eigenvalues and underestimated small eigenvalues. In both cases, the goal is to smooth the spectrum to make it more isotropic, thus an easy target matrix to start with is the scaled identity matrix, which is noted as $T = \alpha I$, where $\alpha$ is the scaling parameter and $I$ is the identity matrix. Thus, the resulting matrix $Y$ and the lower bound in Eq. 2 become

$$Y = (\alpha I)^{1/2} \left( (\alpha I)^{-1/2} K' (\alpha I)^{-1/2} \right)^\beta (\alpha I)^{1/2} = \alpha^{1-\beta} K'^\beta \tag{6}$$

$$\log \rho(K, Y) \geq \mathcal{L}(\beta) = \tfrac{1}{d} \sum_{i=1}^k \log \lambda_{\sigma_i} + \tfrac{1}{d} \sum_{i=1}^d \log \lambda_{\nu_i}^\beta - \tfrac{1}{2} \log \sum_{i=1}^k \lambda_{\sigma_i}^2 - \tfrac{1}{2} \log \sum_{i=1}^d \lambda_{\nu_i}^{2\beta} \tag{7}$$

It is easy to show that, when $\beta = 0$, the resulting matrix $Y$ becomes the target matrix $\alpha I$, and when $\beta = 1$, no shrinkage is performed as $Y = K'$. Eq. 7 indicates that the lower bound is only a function of $\beta$ with no involvement from the scaling parameter $\alpha$ brought by the target matrix as the CKA is invariant to isotropic scaling.

### 3.4 Noiseless Estimation

Starting from the simplest case, we assume that the estimation process from learning word vectors $E$ to constructing the estimated gram matrix $K'$ is noiseless, thus the first order and the second order derivative of $\mathcal{L}(\beta)$ with respect to $\beta$ are crucial for finding the optimal $\beta^\star$:

$$\frac{\partial \mathcal{L}(\beta)}{\partial \beta} = \sum_{i=1}^d \frac{1}{d} \log \lambda_{\nu_i} - \sum_{i=1}^d \frac{\lambda_{\nu_i}^{2\beta}}{\sum_{j=1}^d \lambda_{\nu_j}^{2\beta}} \log \lambda_{\nu_i} \tag{8}$$

$$\frac{\partial^2 \mathcal{L}(\beta)}{\partial \beta^2} = -2 \sum_{i=1}^d \frac{\lambda_{\nu_i}^{2\beta} \log^2 \lambda_{\nu_i}}{\sum_{i=1}^d \lambda_{\nu_i}^{2\beta}} + 2 \frac{\left( \sum_{i=1}^d \lambda_{\nu_i}^{2\beta} \log \lambda_{\nu_i} \right)^2}{\left( \sum_{i=1}^d \lambda_{\nu_i}^{2\beta} \right)^2} \tag{9}$$

Since $\mathcal{L}(\beta)$ is invariant to isotropic scaling on $\{\lambda_{\nu_i} | i \in 1, 2, ..., d\}$, we are able to scale them by the inverse of their sum, then $p_{\nu_i} = \lambda_{\nu_i} / \sum_{i=1}^d \lambda_{\nu_i}$ defines a distribution. To avoid losing information of the estimated gram matrix $K'$, we redefine the plausible range for $\beta$ as $(0, 1]$. Eq. 8 can be interpreted as

$$\frac{\partial \mathcal{L}(\beta)}{\partial \beta} = \sum_{i=1}^d q_i \log p_{\nu_i} - \sum_{i=1}^d r(\beta)_i \log p_{\nu_i} \qquad (q_i = \tfrac{1}{d}, r(\beta)_i = \tfrac{\lambda_{\nu_i}^{2\beta}}{\sum_{j=1}^d \lambda_{\nu_j}^{2\beta}})$$

$$= \tfrac{1}{2\beta} (H(r(\beta)) - H(q, r(\beta))) = \tfrac{1}{2\beta} (H(r(\beta)) - H(q) - D_{KL}(q||r(\beta))) < 0 \tag{10}$$

where $D_{KL}(q||p)$ is the Kullback-Leibler divergence and it is non-negative, $H(q)$ is the entropy of the uniform distribution $q$, and $H(r(\beta), p)$ is the cross-entropy for $r$ and $p$. The inequality derives from the fact that the uniform distribution has the maximum entropy. Thus, the first order derivative $\mathcal{L}'(\beta)$ is always less than 0. With the same definition of $q_i$ and $r(\beta)_i$ in Eq. 10, the second order derivative $\mathcal{L}''(\beta)$ can be rewritten as

$$\frac{\partial^2 \mathcal{L}(\beta)}{\partial \beta^2} = \sum_{i=1}^d r(\beta)_i \log p_{\nu_i} \left( \sum_{j=1}^d r(\beta)_i \log p_{\nu_j} - \sum_{k=1}^d q_k \log p_{\nu_k} \right)$$

$$= \tfrac{1}{2\beta} H(r(\beta), p) (H(r(\beta)) - H(q, r(\beta))) < 0 \tag{11}$$

As shown, $\mathcal{L}'(\beta) < 0$ means that $\mathcal{L}(\beta)$ monotonically decreases in the range of $(0, 1]$ for $\beta$, and the resulting matrix $Y = \alpha I$ completely ignores the estimated gram matrix $K'$. $\mathcal{L}'(\beta)$ also monotonically decreases in the range of $(0, 1]$ for $\beta$ because $\mathcal{L}''(\beta) < 0$.

The observation is similar to the finding reported in previous work (Schäfer & Strimmer, 2005) on shrinkage estimation of covariance matrices that any target will increase the similarity between the oracle covariance matrix and the mixed covariance matrix. In our case, simply reducing $\beta$ from 1 to 0 increases Eq. 2, and consequently, $\beta \to 0_+$ loses all information estimated which is not ideal.

However, one can rewrite Eq. 11 as $\mathcal{L}''(\beta) = H(r(\beta), p)^2 - H(q)H(r(\beta), p)$ and $q$ is the uniform distribution which has maximum entropy, the derivative of $\mathcal{L}''(\beta)$ with respect to $H(r(\beta), p)$ gives

$2H\left(r(\beta),p\right)-H(q)$, and by setting the derivative to $0$, it tells us that there exists a $\beta$ which results in a specific $r$ that gives $H\left(r(\beta),p\right)=\frac{1}{2}H(q)$. Since $H\left(r(\beta),p\right)$ monotonically increases when $\beta$ moves from $0_+$ to $1$, and $\lim_{\beta\to 0_+}H\left(r(\beta),p\right)=H\left(q,p\right)<\frac{1}{2}H(q)$ and $H\left(r(1),p\right)>\frac{1}{2}H(q)$, there exists a single $\beta\in(0,1]$ that gives the smallest value of $\mathcal{L}''(\beta)$. This indicates that there exists a $\hat{\beta}$ that leads to the slowest change in the function value $\mathcal{L}(\beta)$. Then, we simply set $\beta^\star=\hat{\beta}$. In practice, $\beta$ should be larger than $0.5$ in order not to make the resulting matrix overpowered by the target matrix, so one could run a binary search on $\beta\in[0.5,1]$ that gives smallest value of $\mathcal{L}''(\beta)$.

Intuitively, this method is similar to using an elbow plot to find the optimal number of components to keep when running Principal Component Analysis (PCA) on observed data. To translate in our case, a larger value of $\beta$ leads to flatter spectrum and higher noise level as the scaled identity matrix represents isotropic noise. The optimal $\beta^\star$ gives us a point where the increase of $\mathcal{L}(\beta)$ is the slowest.

Once optimal mixing parameters $\beta^\star$ are found, the resulting matrix $\boldsymbol{Y}=\boldsymbol{K}'^{\beta^\star}=\boldsymbol{E}^{\beta^\star}(\boldsymbol{E}^{\beta^\star})^\top$, and $\alpha$ is omitted here. As a post-processing method for word vectors, we propose to transform the word vectors in the following fashion:

1. $\boldsymbol{E}=\boldsymbol{E}-\frac{1}{n}\sum_{i=1}^n\boldsymbol{E}_{i\cdot}$
2. Singular Value Decomposition $\boldsymbol{E}=\boldsymbol{U}\boldsymbol{S}\boldsymbol{V}^\top$
3. Optimal Mixing Parameter $\beta^\star=\arg\min_\beta\mathcal{L}''$
4. Reconstruct the matrix $\boldsymbol{E}^\star=\boldsymbol{U}(\boldsymbol{S})^{\beta^\star}\boldsymbol{V}^\top$

## 4 EXPERIMENTS

Three learning algorithms are applied to derive word vectors, including skipgram (Mikolov et al., 2013b), CBOW (Mikolov et al., 2013a) and GloVe (Pennington et al., 2014).[2] $\boldsymbol{E}\boldsymbol{E}^\top$ serves as the estimated gram matrix $\boldsymbol{K}'$. Hyperparameters of each algorithm are set to the recommended values. The training corpus is collected from Wikipedia and it contains around 4600 million tokens. The unique tokens include ones that occur at least 100 times in the corpus and the resulting dictionary size is 366990. After learning, the word embedding matrix is centralised and then the optimal mixing parameter $\beta^\star$ is directly estimated from the singular values of the centralised word vectors. Afterwards, the postprocessed word vectors are evaluated on a series of word-level tasks to demonstrate the effectiveness of our method. Although the range of $\beta$ is set to $[0.5,1.0]$ and the optimal value is found by minimising $\mathcal{L}''(\beta)$, it seldomly hits $0.5$, which means that searching is still necessary.

### 4.1 COMPARISON PARTNERS

Two comparison partners are chosen to demonstrate the effectiveness of our proposed method. One is the method (Mu et al., 2018) that removes the top 2 or 3 principal components on the zero-centred word vectors, and the number of top principal components to remove depends on the training algorithm used for learning word vectors, and possibly the training corpus. Thus, the selection process is not automatic, while ours is. The hypothesis behind this method is similar to ours, as the goal is to make the learnt word vectors more isotropic. However, removing top principal components, which are correlated with the frequency of each word, could also remove relevant information that indicates the relationship between words. The observation that our method is able to perform on par with or better than this comparison partner supports our hypothesis.

The other comparison partner is the optimal shrinkage (Ledoit & Wolf, 2004) on the estimated covariance matrices as the estimation process tends to overestimate the directions with larger power and underestimate the ones with smaller power, and the optimal shrinkage aims to best recover the true covariance matrix with minimum assumptions about it, which has the same goal as our method. The resulting matrix $\boldsymbol{Y}$ is defined to be a linear combination of the estimated covariance matrix and the target with a predefined structure, and it is denoted as $\boldsymbol{Y}=(1-\beta)\alpha\boldsymbol{I}+\beta\boldsymbol{\Sigma}'$, where in our case, $\boldsymbol{\Sigma}'=\boldsymbol{E}^\top\boldsymbol{E}\in\mathbb{R}^{d\times d}$. The optimal shrinkage parameter $\beta$ is found by minimising the Euclidean distance between the true covariance matrix $\boldsymbol{\Sigma}$ and the resulting matrix $\boldsymbol{Y}$. The issue here is that Euclidean distance and linear combination may not be the optimal choice for covariance matrices, and also Euclidean distance is not invariant to isotropic scaling which is a desired property when measuring the similarity between sets of word vectors.

---

[2]https://github.com/facebookresearch/fastText, https://github.com/stanfordnlp/GloVe

## 4.2 Word-level Evaluation Tasks

Three categories of word-level evaluation tasks are considered, including 8 tasks for word similarity, 3 for word analogy and 6 for concept categorisation. The macro-averaged results for each category of tasks are presented in Table 1. Details about tasks are included in the appendix.

Table 1: **Results of three post-processing methods including ours on our pretrained word vectors from three learning algorithms.** In each cell, the three numbers refer to word vectors produced by "Skipgram / CBOW / GloVe", and the dimension of them is 500. **Bold** indicates the best macro-averaged performance of post-processing methods. It shows that overall, our method is effective.

| Postprocessing | Similarity (8) | Analogy (3) | Concept (6) | **Overall** (17) |
|---|---|---|---|---|
| None | **64.66** / 57.65 / 54.81 | 46.23 / 48.70 / 46.68 | 68.70 / 68.34 / 66.86 | 62.83 / 59.84 / 57.63 |
| Mu et al. (2018) | 64.24 / 60.12 / **61.57** | **46.87** / 49.10 / 40.09 | 68.86 / **70.46** / 65.54 | 62.81 / 61.83 / **59.18** |
| Ledoit & Wolf (2004) | 63.39 / 59.19 / 50.81 | 46.64 / **50.38** / 45.54 | 69.45 / 68.83 / **69.55** | 62.57 / 61.03 / 56.50 |
| Ours | 63.79 / **60.90** / 53.16 | 46.51 / 49.84 / **47.25** | **70.83** / 69.84 / 65.97 | **63.23** / **62.10** / 56.54 |

## 4.3 Word Translation Tasks

Table 2 presents results on supervised and unsupervised word translation tasks (Lample et al., 2018)[3] given pretrained word vectors from FastText (Bojanowski et al., 2017). The supervised word translation is formalised as solving a Procrustes problem directly. The unsupervised task first trains a mapping through adversarial learning and then refines the learnt mapping through the Procrustes problem with a dictionary constructed from the mapping learnt in the first step. The evaluation is done by k-NN search and reported in top-1 precision.

Table 2: **Performance on word translation.** The translation is done through k-NN with two distances, in which one is the cosine similarity (noted as "NN" in the table), and the other one is the Cross-domain Similarity Local Scaling (CSLS) (Lample et al., 2018). The Ledoit & Wolf's method didn't converge on unsupervised training so we excluded results from the method in the table.

| Post-processing | en-es | | es-en | | en-fr | | fr-en | | en-de | | de-en | |
|---|---|---|---|---|---|---|---|---|---|---|---|---|
| | NN | CSLS | NN | CSLS | NN | CSLS | NN | CSLS | NN | CSLS | NN | CSLS |
| **Supervised Procrustes Problem** | | | | | | | | | | | | |
| None | 79.0 | 81.7 | 79.2 | **83.3** | 78.4 | 82.1 | 78.5 | 81.9 | 71.1 | 73.4 | 69.7 | 72.7 |
| Mu et al. (2018) | 80.0 | 81.7 | 80.4 | 83.0 | **79.3** | **82.4** | 78.5 | 81.7 | **72.9** | **75.4** | 70.7 | 73.1 |
| Ledoit & Wolf (2004) | 80.1 | 82.3 | 79.6 | 82.9 | 78.7 | 81.9 | 78.9 | **82.0** | 72.3 | 74.1 | 70.6 | 72.4 |
| Ours | **81.1** | 82.3 | **80.9** | 83.0 | **79.3** | 82.1 | **79.2** | 81.7 | 72.5 | 74.5 | **71.7** | **72.9** |
| **Unsupervised Adversarial Training + Refinement** | | | | | | | | | | | | |
| None | 79.7 | 82.0 | 78.8 | 83.7 | 78.4 | 81.8 | 77.9 | **82.1** | 71.7 | 74.7 | 69.7 | 73.1 |
| Mu et al. (2018) | 79.7 | 81.7 | 80.5 | **84.3** | **79.6** | **82.7** | 78.9 | 81.8 | **72.3** | **75.0** | 71.2 | 72.9 |
| Ours | **80.4** | **82.3** | **81.1** | 83.1 | 79.5 | 82.0 | **79.5** | **82.1** | 72.1 | 74.8 | **71.7** | **73.7** |

## 4.4 Sentence-level Evaluation Tasks

The evaluation tasks include five tasks from SemEval Semantic Textual Similarity (STS) in 2012-2016 (Agirre et al., 2012; 2013; 2014; 2015; 2016). A sentence vector is constructed by simply averaging word vectors or postprocessed word vectors, and the similarity measure is cosine similarity as well. The performance on each task is reported as the Pearson's correlation score.

We evaluate our post-processing at two different levels, in which one applies the post-processing methods on word vectors before averaging them to form sentence vectors (Mu et al., 2018), and the other one applies methods on averaged word vectors, which are sentence vectors, directly (Arora

---

[3]https://github.com/facebookresearch/MUSE

et al., 2017). Applying the post-processing to sentence vectors results in better performance and our method leads to the best performance on each dataset. Results are reported in Table 3.

Table 3: **Performance of our post-processing method and the other two comparison partners on SemEval datasets.** The task is to make good predictions of sentence vectors composed of averaging word vector. The **word-level** post-processing means that all methods are applied on word vectors before averaging, and the **sentence-level** one means that all methods are applied on sentence vectors. Our method performs better than others at sentence-level and slightly worse than the method that removes top principal components on the word level. Overall the sentence-level post-processing results in superior performance. Our method has the best overall performance on each dataset.

| Postprocessing | STS12 | STS13 | STS14 | STS15 | STS16 | STS12 | STS13 | STS14 | STS15 | STS16 |
|---|---|---|---|---|---|---|---|---|---|---|
| | word-level post-processing (Mu et al., 2018) | | | | | sentence-level post-processing (Arora et al., 2017) | | | | |
| RM Top PCs | **59.07** | **53.30** | **65.98** | **69.54** | **67.22** | 59.20 | 57.92 | 69.31 | 74.08 | 72.26 |
| Ledoit & Wolf (2004) | 54.75 | 47.93 | 60.57 | 65.36 | 64.87 | 54.47 | 57.92 | 64.79 | 70.84 | 58.53 |
| Ours | 57.36 | 51.80 | 65.04 | 68.23 | 66.31 | **62.60** | **62.09** | **70.99** | **75.83** | **74.49** |

## 5 DISCUSSION

The results presented in Table 1, 2 and 3 indicate the effectiveness of our methods. In addition, by comparing to two most related post-processing methods, which are removing top principal components (Mu et al., 2018; Arora et al., 2017) and the optimal shrinkage estimation on covariance matrices in Euclidean space (Ledoit & Wolf, 2004; Schäfer & Strimmer, 2005), our method combines the best of the both worlds - the good performance of the first method and the automaticity property of the second one.

Specifically, our method derives the resulting gram matrix in a semi-Riemannian manifold, and the shrinkage is done by moving the estimated gram matrix $K'$ to the scaled identity matrix $\alpha I$ on the geodesic on the semi-Riemannian manifold defined in Eq. 6. Compared to measuring the Euclidean distance between two covariance matrices (Ledoit & Wolf, 2004), we chose a distance measure that respects the structure of the space where positive semi-definite matrices exist. Also, since we are working with improving the quality of learnt word vectors, given that mostly the angular distance between a pair of word vectors matters, the similarity measure between two spaces needs to be invariant to rotation and isotropic scaling, and the Centralised Kernel Alignment is a good fit in our case. Compared to the post-processing method that removes top principal components, our method defines an objective function where the unique minimum point in the specific range indicates the optimal shrinkage parameter $\beta^\star$, and which doesn't require any human supervision on finding the parameters (the number of principal components to remove in that method).

The derivations in our work are mostly done on the lower bound in Eq. 2 of the original log of CKA $\rho(K, K')$, and the bound is tight when $KK'$ has a flat spectrum. In this sense, the lower bound essentially defines the similarity of the resulting matrix $Y$ and a scaled identity matrix where the scaling factor is the average of the eigenvalues of $KK'$. As the isotropic scaling can be ignored in our case, intuitively, the lower bound (Eq. 2) gives us how to shrink an estimated gram matrix to an identity matrix. Since the target matrix $T = \alpha I$ is also a scaled identity matrix, the lower bound keeps increasing as $Y$ travels from $K'$ to $T$, and it explains why the first order derivative is negative and the second order derivative is more informative. Our method is very similar to using an elbow plot to find the number of eigenvalues to keep in PCA. Future work should focus on finding a tighter bound that considers the interaction between the true gram matrix $K$ and the estimated one $K'$.

Figure 1 shows that our methods give solid and stable improvement against two comparison partners on learnt word vectors provided by skipgram and CBOW with different dimensions, and slightly worse than removing top PCs on GloVe. However, skipgram and CBOW provide much better performance thus improvement on these algorithms is more meaningful, and our method doesn't require manual speculations on the word vectors to determine the number of PCs to remove. An interesting observation is that there is a limit on the performance improvement from increasing the dimensionality of word vectors, which was hypothesised and analysed in prior work (Yin & Shen, 2018). Figure 2 shows that our method consistently outperforms two comparison partners when limited amount of

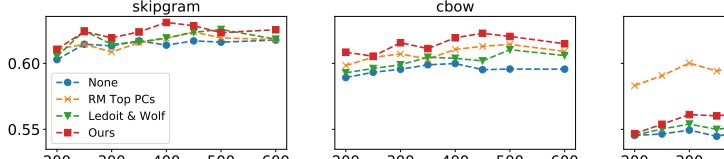

Figure 1: **Performance change of each post-processing method when the dimension of learnt word vectors increases.** Our method is comparable to the method that removes top PCs on skipgram, and better than that on CBOW, and slightly worse than that on GloVe. However, generally Skipgram and CBOW themselves provide better performance, thus boosting performance by postprocessing on those two learning algorithms is more important. In addition, our model doesn't require manually picking the dimensions to remove instead the optimal $\beta^\star$ is found by optimisation.

training corpus is provided, and the observation is consistent across learnt word vectors with varying dimensions. From the perspective of shrinkage estimation of covariance matrices, the main goal is to optimally recover the oracle covariance matrix under a certain assumption about the oracle one when limited data is provided. The results presented in Figure 2 indicate that our method is better at recovering the gram matrix of words, where relationships of word pairs are expressed, as it gives better performance on word-level evaluation tasks on average.

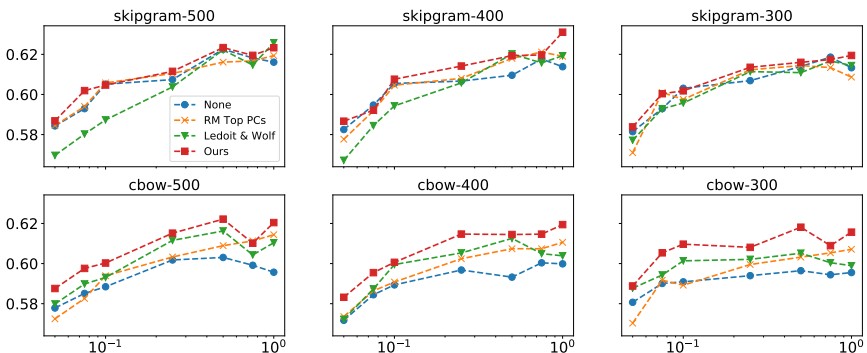

Figure 2: **Averaged score on 17 tasks vs. Percentage of training data.** To simulate the situation where only limited data is available for shrinkage estimation, two algorithms are trained with subsampled data with different portions, and three postprocessing methods including ours and two comparison partners are applied afterwards. The observation here is that our method is able to recover the similarity between words better than others when only small amount of data is available.

## 6 CONCLUSION

We define a post-processing method in the view of shrinkage estimation of the gram matrix. Armed with CKA and geodesic measured on the semi-Riemannian manifold, a meaningful lower bound is derived and the second order derivative of the lower bound gives the optimal shrinkage parameter $\beta^\star$ to smooth the spectrum of the estimated gram matrix which is directly calculated by pretrained word vectors. Experiments on the word similarity and sentence similarity tasks demonstrate the effectiveness of our model. Compared to the two most relevant post-processing methods (Mu et al., 2018; Ledoit & Wolf, 2004), ours is more general and automatic, and gives solid performance improvement. As the derivation in our paper is based on the general shrinkage estimation of positive semi-definite matrices, it is potentially useful and beneficial for other research fields when Euclidean measures are not suitable. Future work could expand our work into a more general setting for shrinkage estimation as currently the target in our case is an isotropic covariance matrix.

## ACKNOWLEDGEMENTS

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

APPENDIX

## A  WORD-LEVEL EVALUATION TASKS

**8 word similarity tasks**: MEN(Bruni et al., 2014), SimLex999(Hill et al., 2015), MTurk(Halawi et al., 2012), WordSim353(Finkelstein et al., 2002), WordSim-353-REL(Agirre et al., 2009), WordSim-353-SIM(Agirre et al., 2009), RG65(Rubenstein & Goodenough, 1965). On these, the similarity of a pair of words is calculated by the cosine similarity between two word vectors, and the performance is reported by computing Spearman's correlation between human annotations with the predictions from the learnt vectors.

**3 word analogy tasks**: Google Analogy(Mikolov et al., 2013a), SemEval-2012(Jurgens et al., 2012), MSR(Mikolov et al., 2013c). The task here is to answer the questions in the form of "$a$ is to $a^\star$ as $b$ is to $b^\star$", where $a$, $a^\star$ and $b$ are given and the goal is to find the correct answer $b^\star$. The "3CosAdd" method (Levy et al., 2015) is applied to retrieve the answer, which is denoted as $\arg\max_{b^\star \in V_w \setminus \{a^\star, b, a\}} = \cos(b^\star, a^\star - a + b)$. Accuracy is reported.

**6 concept categorisation tasks**: BM(Baroni et al., 2010), AP(Almuhareb, 2006), BLESS(Jastrzebski et al., 2017), ESSLI(Baroni et al., 2008). The task is to measure whether words with similar meanings are nicely clustered together. An unsupervised clustering algorithm with predefined number of clusters is performed on each dataset, and the performance is reported also in accuracy.

## B  PYTHON CODE

```python
import numpy as np

def beta_postprocessing(emb):
    # embedding size: number of word vectors x dim of vectors

    mean = np.mean(vecs, axis=0, keepdims=True)
    centred_emb = emb - mean
    u, s, vh = np.linalg.svd(centred_emb)

    def objective(b):
        l = s ** 2
        logl = np.log(l)
        l2b = l ** (2*b)

        first_term = np.mean(logl*l2b*logl)*np.mean(l2b)
        second_term = np.mean(logl*l2b)**2.
        derivative = - 1. / (np.mean(l2b) ** 2.) * (first_term - second_term)
        return derivative

    values = list(map(objective, np.arange(0.5, 1.0, 0.001)))
    optimal_beta = np.argmin(values) / 1000. + 0.5

    processed_emb = u @ np.diag(s ** optimal_beta) @ vh

    return processed_emb
```

Listing 1: Python code for searching for the optimal $\beta^\star$

