# OpenReview forum: "An Empirical Study on Post-processing Methods for Word Embeddings"
_ICLR.cc/2020/Conference — Reject_

### Official Review · AnonReviewer2 · 2019-10-19
**Official Blind Review #2**

**Rating:** 1

**Review:**

This paper proposes a theoretical framework for post-processing methods of word embeddings. Given the framework, the authors derive their own methods and then a thorough experimental analysis follows.

While the paper reflects thorough and substantial work - both in the theoretical framework and in the experimental part, I have serious concerns about its clarity and about the experimental results and also some concerns about comparison with previous work. All these, unfortunately, make me recommend a reject decision. Below, are more details:

1. Clarity: There are multiple aspects of the paper that I needed to read a few times before I understood what the authors mean.

First, the problem definition is not clear: Even when I reached the end of the introduction, it was not clear to me what is the problem that the authors are trying to solve and what they are trying to achieve. One example of this problem is the title of the paper, which suggests that the author performs an empirical evaluation, while in practice they also suggest their own method. But this is a much broader issue - the author suggest very little motivation to how the post-processing method should work, what improvements it should provide and why we should expect such improvements. At some points I felt that this is more of a clean mathematical exercise than a discussion of methods that should improve word representations in natural language.

In addition, the authors assume a strong background in a very specific post-processing  literature and do not provide any details about its fundamentals. Only on the beginning of section 3 I learned the fundamentals of that framework and basic concepts such as the Gram matrix. I believe a scientific paper should be self-contained, the motivations, goals and fundamental concepts form previous work should be clearly stated and explained. This is not done in this paper, unfortunately.

Finally, it was hard for me to determine where the survey of previous work ends and the contribution of this work begins. Particularly, it seems that the authors propose a unified framework for the methods in previous work and it is not clear which parts of that framework were already discussed in previous work and which are original contributions of the authors. This makes it also hard to estimate how different the proposed method is form the previous ones.

2. Experimental analysis:

First, the authors describe their evaluation tasks very briefly and only in the appendix. This is just a list of tasks with no insight about natural language (please see a related comment in the clarity section of this review). Then, the results are reported as a macro-average over many tasks: Given the large number of tasks this is a very crude average, and there is no way that any real insight into the change/improvement of the vectors can be derived from this report. Finally, the reported numbers reflect very minor improvements, if at all, compared to previous post-processing methods and to the original vectors. Again, since these are macro-averages over many tasks, the conclusions that can be derived are very limited (e.g. it might be that the proposed method does improve on some of the tasks and harm the performance on others, or that it keeps the vectors very similar to the original ones - we have no way figuring out the actual picture).


3. Comparison to previous work:

As said above, it seems that the authors view their work in a narrow context of a very specific literature. In fact, the NLP literature contains a large number of post-processing word embedding methods (often referred to as "fitting" or "specialization" methods). While these methods sometimes build on external linguistic knowledge (e.g. from wordNet or from other manually crafted lexicons), they have also shown useful with automatically constructed constraints, that are similar to the structural considerations mentioned in the paper in the sense that they do not require expert knowledge, they only build on common-sensical requirements from a good vector space for word meaning representation. Some relevant papers are:

Faruqui, M., Dodge, J., Jauhar, S. K., Dyer, C., Hovy, E., & Smith, N. A. (2014). Retrofitting word vectors to semantic lexicons. arXiv preprint arXiv:1411.4166.

Mrkšić, N., Séaghdha, D. O., Thomson, B., Gašić, M., Rojas-Barahona, L., Su, P. H., ... & Young, S. (2016). Counter-fitting word vectors to linguistic constraints. arXiv preprint arXiv:1603.00892.‏‏

Mrkšić, N., Vulić, I., Ó Séaghdha, D., Leviant, I., Reichart, R., Gašić, M., ... & Young, S. (2017). Semantic specialization of distributional word vector spaces using monolingual and cross-lingual constraints. Transactions of the association for Computational Linguistics, 5, 309-324.‏

**Experience Assessment:**

I have published in this field for several years.

**Review Assessment: Checking Correctness Of Derivations And Theory:**

I assessed the sensibility of the derivations and theory.

**Review Assessment: Checking Correctness Of Experiments:**

I carefully checked the experiments.

**Review Assessment: Thoroughness In Paper Reading:**

I read the paper thoroughly.

---

### Official Review · AnonReviewer1 · 2019-10-23
**Official Blind Review #1**

**Rating:** 6

**Review:**

The authors propose a novel word embedding post-processing method that maximizes the similarity between the estimated Gram matrix of word vectors and its oracle matrix. To find the optimal Gram matrix, they adopt the shrink method to make Gram matrix K' to target matrix T on semi-Riemannian space. The authors use the shrinkage method to find optimal K' and formulate the maximization on CKA problem as finding the optimal shrinkage parameter that maximizes the lower bound of the CKA between the estimated Gram matrix and oracle. By applying the proposed method to various word embedding methods, the authors show the performance of their post-processing method on word analogy/similarity task, word translation task, and sentence similarity task.

Strengths
* This paper provides a novel post-processing method that can relieve isotropy condition and shows experimental support that solving isotropy condition on word embedding vectors can improve its performance.
* Large set of experiments on various word embedding benchmark tasks.

Weaknesses
* It would be nice if the authors show the performance of the post-processed word vectors on other NLP benchmark: text classification, NER, ... etc.

Question
* "Neural Word Embedding as Implicit Matrix Factorization" and "Analogies Explained: Towards Understanding Word Embeddings" show that PMI is the global optimum point of the previous word embedding model's problem space and prove word analogy can be explained from the PMI characteristics of word embedding models. How can this paper be related to the two papers mentioned above?

**Experience Assessment:**

I have read many papers in this area.

**Review Assessment: Checking Correctness Of Derivations And Theory:**

I assessed the sensibility of the derivations and theory.

**Review Assessment: Checking Correctness Of Experiments:**

I carefully checked the experiments.

**Review Assessment: Thoroughness In Paper Reading:**

I read the paper at least twice and used my best judgement in assessing the paper.

---

### Official Review · AnonReviewer3 · 2019-10-25
**Official Blind Review #3**

**Rating:** 3

**Review:**

This paper presents a novel method for unsupervised post-processing of pretrained word embeddings that enforces the distributional word vector space to be more isotropic, which in turn improves the expressiveness and quality of the space in terms of similarity. The method is based on the shrinkage of the covariance/Gram matrix and its effects on the input space are evaluated across a range of intrinsic evaluation tasks. While I like the idea overall and this line of work in general, there are still some concerns with the current version of the paper:

* Overall, although its design seems more principled, the proposed method does not seem significantly better than the previous (very similar) method of Mu et al. I would like to see more evidence in the favour of the proposed method, pointing that we should use that instead of Mu's method. Also, the gains over non-processed spaces often seem insignificant, and offer only small benefits.

* The derivation of the method seems too verbose, especially in light of the fact that it is directly inspired by previous work on the shrinkage estimation of covariance matrices and CKA. I would suggest the authors to spend more time on linking their high-level hypotheses to the low-level mathematical implementations instead of flooding the paper with equations - for the interested reader a lot of the derivation process can be put into an appendix, the paper should focus on conveying the key principles instead. This would also offer additional space for more experiments.

* One very relevant paper is not mentioned at all: https://arxiv.org/pdf/1809.02094.pdf (Artetxe et al., CoNLL 2018). I would suggest the authors to cite that work and ideally even compare to it on their set of intrinsic tasks (e.g., word similarity, word analogy), and then discuss the difference in results and their approach to unsupervised post-processing. This shouldn't be so difficult to do as the code from that paper is available online: https://github.com/artetxem/uncovec

* The paper mixes true similarity datasets (such as SimLex) with broader semantic relatedness datasets (MEN, WordSim-353), even mixing true semantic similarity of the same dataset (WordSim353-SIM) with its relatedness subset (WordSim353-REL). In light of the known conceptual differences between the relations of similarity versus relatedness, I would suggest to report the results separately for the two tasks. For instance, another true similarity dataset, which is not used in the evaluation is SimVerb-3500. Along the same line, it is also not clear what type of similarity is meant when the authors state that through post-processing 'the similarity between words can be better expressed'. What does it mean to better express the similarity between words in the first place? Do we talk about true similarity or relatedness or both? However, the two relationships between words support different classes of downstream applications, so therefore it is even more problematic 1) not to distinguish between the two and 2) not to report any results in any downstream (extrinsic) tasks where the post-processed embeddings are used as features (STS is still a semi-intrinsic task imho; BLI is considered as an intrinsic task in cross-lingual settings).

* One of the key reasons to apply post-processing is to mitigate the frequency artefacts: however, such an evaluation that goes towards that direction is never executed. For instance, I would like to see a focused experiment that measures how post-processing affects high-frequency versus mid-to-lower frequency or rare words. A recently developed CARD-660 dataset might be used to this end.

* It is not clear how exactly the authors run Mu et al.'s method, that is, how many top principal components are removed for the input vectors? How is this selected? Are always the optimal results reported for the baseline Mu et al.'s method? Why not reporting the results with removing 2 and 3 at the same time to further prove the point that their method is non-automatic? This would also give the reader a hint how much the results with Mu's method actually differ/decrease if one just decides to make the method 'automatic' by just fixing the method to always remove the same number of top principal components.


Minor remarks:
* The title of the paper is a bit imprecise: in the word embedding literature, the term post-processing is often referred to the methods that fine-tune word embeddings using some external knowledge after (i.e., post) the initial distributional training (e.g., the so-called retrofitting methods). However, in the context of the paper post-processing actually refers to some unsupervised post-training steps on the input space without injecting any external information. This should be made clearer in the paper, and perhaps adding a paragraph which outlines the core difference to other work on retrofitting would be helpful as well.

* I might be missing something while reading Section 3, but it is currently not clear to me how the oracle Gram matrix K is obtained in the first place. Perhaps it makes sense to briefly summarize this in a quite direct way to avoid the reader's confusion?

* It is great to see a summary of the key post-processing steps at the very end of Section 3; this is really helpful for everyone who would like to try out the proposed method off-the-shelf. However, the summary is not self-contained as it is not clear what \mathcal{L}'' refers to (and the reader must search through the derivations again to find its meaning).

* I like the evaluation on word translation, and I believe that the proposed post-processing methods could actually improve word translation through some pre-alignment perturbations. It is a pity that the method is not evaluated on more distant language pairs, as I believe that the method might have much more effect there than on the already-saturated EN-to-ES/FR/IT bilingual lexicon induction tasks.

* For de-en word translation Mu's method actually beats the proposed method (incorrect number in bold)

* It is not clear why MUSE is used for word retrieval experiments, given the fact that it is known to be unstable (Sogaard et al., ACL 2019), and there are more robust and more effective methods available such as VecMap (Artetxe et al., ACL 2018) or RCSLS (Joulin et al., EMNLP 2019)

**Experience Assessment:**

I have published one or two papers in this area.

**Review Assessment: Checking Correctness Of Derivations And Theory:**

I assessed the sensibility of the derivations and theory.

**Review Assessment: Checking Correctness Of Experiments:**

I carefully checked the experiments.

**Review Assessment: Thoroughness In Paper Reading:**

I read the paper at least twice and used my best judgement in assessing the paper.

---

### Public Comment · ~Mozhi_Zhang1 · 2019-11-07
**Question about word translation and possibly related work**

Dear authors,

For the word translation task, I wonder whether the post-processing happens before the alignment (i.e., it is applied to the two monolingual embeddings) or after the alignment (i.e., it is applied to the shared embedding space).

This work reminds me of a related paper: https://arxiv.org/pdf/1906.01622.pdf , where the authors use alternating projection to normalize monolingual embeddings to make them easier to align, and it empirically improves translation accuracy for Procrustes. I would be interesting to see comparison between the proposed method and this previous work (the code is here: https://gist.github.com/zhangmozhi/1e37c997514115e9b63476e322ca2ad0 ).

It would also be interesting to see experiments on more language pairs. For some more distant language pairs, the post-processing method may be even more useful.

---

### Author Response · Authors · 2019-11-15
**Thanks for the constructive comments!**

We really appreciate reviewers' and audience's comments on improving our paper, and we uploaded a new version of our submission. Thanks,

---

### Decision · Program_Chairs · 2019-12-19

**Decision:**

Reject

**Comment:**

This paper explores a post-processing method for word vectors to "smooth the spectrum," and show improvements on some downstream tasks.

Reviewers had some questions about the strength of the results, and the results on words of differing frequency. The reviewers also have comments on the clarity of the paper, as well as the exposition of some of the methods.

Also, for future submissions to ICLR and other such conferences, it is more typical to address the authors comments in a direct response rather than to make changes to the document without summarizing and pointing reviewers to these changes. Without direction about what was changed or where to look, there is a lot of burden being placed on the reviewers to find your responses to their comments.